# Morphological Evaluation of Transscleral Laser Retinopexy in Rabbits: Comparison of Optical Coherence Tomography and Histologic Examinations

**DOI:** 10.3390/vetsci10090535

**Published:** 2023-08-23

**Authors:** Maria Vanore, Tristan Juette, Javier Benito, Marie-Odile Benoit-Biancamano

**Affiliations:** 1Centre Hospitalier Universitaire Vétérinaire, Département de Sciences Cliniques, Faculté de Médecine Vétérinaire, Université de Montréal, 3200 Rue Sicotte, Saint-Hyacinthe, QC J2S 2M2, Canada; javier.benito@umontreal.ca; 2Faculté de Médecine Vétérinaire, Université de Montréal, 3200 Rue Sicotte, Saint-Hyacinthe, QC J2S 2M2, Canada; tristan.juette@umontreal.ca; 3Groupe de Recherche sur les Maladies Infectieuses en Production Animale (GREMIP), Département de Pathologie et Microbiologie, Faculté de Médecine Vétérinaire, Université de Montréal, 3200 Rue Sicotte, Saint-Hyacinthe, QC J2S 2M2, Canada; marie-odile.benoit-biancamano@umontreal.ca

**Keywords:** diode laser, histology, optical coherence tomography, photocoagulation, rabbits, transscleral retinopexy

## Abstract

**Simple Summary:**

Transscleral retinopexy is performed to prevent retinal detachment. Retinal lesions induced by the laser have the appearance of retinal burns, (photocoagulation lesions), which evolve into scars (atrophic retinal lesions), inking the retina to the choroid and preventing retinal detachment. It is not known whether this procedure is effective. The current study compared retinal lesions by histology and OCT over time. Retinal damage occurred immediately during transscleral retinopexy. The follow-up of the retinal lesion was carried out 42 days after the laser procedure. The study provided OCT and histologic images that show the morphological evolution of the lesions over time until the scar. The first stage in the photocoagulation lesion is focal edema, which fades on the 7th day. On the 15th, the lesions exhibit tissue fibrosis. On the 21st day, the retinal lesion was excavated and cup-shaped, appearing as a scar. In the future, it will be possible to perform an OCT to identify scars and validate the correct transscleral laser retinopexy outcome.

**Abstract:**

Transscleral retinopexy is a preventive technique used against retinal detachment. Fundus examination can allow the monitoring of morphological retinal changes in the progression of photocoagulation lesions, without offering details on the morphological changes by the retinal lesion. The aim of the study was to assess the progression of photocoagulation lesions induced by transscleral retinopexy (840 nm diode laser), by comparing the optical coherence tomography (OCT) and histological images over a period of six weeks on eight pigmented New Zealand healthy rabbits (four males and four females; n = 16 eyes)**.** All rabbits underwent transscleral retinopexy on their left eye on day 0 (D0). Measurements of the photocoagulation lesions were obtained in vivo on D0, D7, D15, D21, and D42 by acquiring OCT images of both eyes from all rabbits. On D1, D7, D21, and D42, two rabbits were euthanized, and their eyes were enucleated. A significant effect by time on the decrease in the central retinal thickness of the photocoagulation lesion was observed from D1 to D7 (*p* = 0.001); however, no such effect was observed on the horizontal length ((HL) *p* = 0.584) of the lesion surface. The reliability between the OCT and histological measurements, which were evaluated using intraclass correlation coefficients, was excellent for measuring the retinal thickness at the center (ICC = 0.91, *p* < 0.001), moderate for the right side of the retinal lesions (ICC = 0.72, *p* = 0.006), and not significant for the left side and HL (*p* = 0.055 and 0.500, respectively). The morphological changes observed in the OCT and histopathological images of the photocoagulation lesions were qualitatively described over time. OCT is an effective tool for monitoring changes in photocoagulation lesions. Some measurements and qualitative changes showed an adequate correlation between the OCT and histological findings.

## 1. Introduction

Retinal detachment (RD) occurs due to separation in the neurosensory retina from the underlying retinal pigmented epithelium (RPE) [1]. Transscleral retinopexy prevents RD by anchoring the retina to the choroid, even in cases with poor retinal visualization [2,3]. However, to the best of the authors’ knowledge, the progression of retinal changes following transscleral retinopexy has not been described using medical imaging [4,5].

Optical coherence tomography (OCT) is a non-invasive in vivo optical imaging modality, which is capable of providing cross-sectional and volumetric images of the retina in human and veterinary species [6,7]. OCT can be used for in vivo microscopy and in animal experiments, and it can enable real-time in vivo imaging and allow single-subject tracking without sacrificing animals at intermediatory time points [8,9,10,11,12]. OCT acquires high-resolution images of biological microstructures by directing a focused light beam into the biological tissues [13]. Performing transscleral retinopexy with an 810 nm diode laser before cataract surgery reportedly prevents rhegmatogenous retinal detachment in Bichon Frisé [14,15]. Since the effectiveness of transscleral retinopexy is not clear in all cases, OCT may enable the postoperative follow-up of this procedure to assess the treatment outcomes. For this reason, it is relevant to verify whether OCT characterizes the retinal scars formed after diode laser transscleral retinopexy, thereby allowing the monitoring of lesions over time. Therefore, the aim of this study was to qualitatively and quantitatively assess the photocoagulation changes induced in rabbits over the course of six weeks after undergoing 810 nm transscleral diode laser retinopexy and to compare the OCT images by histological sections.

## 2. Materials and Methods

### 2.1. Animals

Eight healthy New Zealand pigmented rabbits (four males and four females), with a mean age of 3.2 ± 0.5 (mean ± SD) months and a mean weight of 2.5 ± 0.2 kg, were enrolled in this study. Animals were purchased from a farm intended for meat production. The rabbit animal model was chosen based on previous studies of transscleral diode photocoagulation [16]. Before inclusion in this study, all rabbits underwent a complete ocular examination to rule out potential ocular diseases. Exclusion criteria included corneal and lens opacity, ocular inflammation, and any retinal lesions. All procedures were conducted in accordance with the Use of Animals in Ophthalmic and Vision Research statement of the Association for Research in Vision and Ophthalmology (ARVO) and the Guidelines for Ethical Conduct in the Care and Use of Animals in Research. The study protocol was approved by the Institutional Animal Care and Use Committee (IACUC-CÉUA 20-Rech-2021).

The rabbits were considered healthy based on the results of physical and complete bilateral ophthalmic examinations. The ophthalmic examination, performed by a board-certified veterinary ophthalmologist, included distance examination, ophthalmic evaluation, slit lamp biomicroscopy (Kowa SL-17^®^; Innova, Toronto, ON, Canada), and indirect ophthalmoscopy using a 3.5 V Finoff transilluminator (Heine, Topcon, Boisbriand, QC, Canada) and 30D and 20D lenses (Volk, Axis Medical, Anjou, QC, Canada). Fundus photography (Kowa Genesis-D, retinal camera fundus) was performed in parallel with OCT to monitor the progression of retinal lesions.

The rabbits were housed in the Division Ferme et Animaleries (FANI) in individual stainless steel cages (120 L × 100 W × 100 H cm) and maintained in an open system. The room temperature (22 ± 2 °C), humidity (63 ± 3%), and animal husbandry were standardized throughout the experiment. The rabbits were acclimated and conditioned for one week before the experiments to allow for physiological adjustment to the environmental changes. The experiments were conducted between 08:00 am and 05:00 pm, using the following methods.

### 2.2. Study Design

This study was designed as a prospective, descriptive, randomized, and blind study. All procedures were performed by board-certified ophthalmologists and/or pathologists.

### 2.3. Transscleral Retinopexy and Ophthalmic Examination

The pupils were dilated by administering a topical solution of 1% tropicamide (Mydriacyl^®^, Novartis, Toronto, ON, Canada), 10 min prior to administering the sedatives.

All rabbits were sedated using an intramuscular (IM) combination of ketamine hydrochloride (6 mg/kg IM, Vetalar^®^; Zoetis, Kirkland, QC, Canada) and hydromorphone (0.15 mg/kg IM, Hydromorphone hydrochloride^®^; Sandoz, Boucherville, QC, Canada) on day 0 (D0). In addition, 0.1 mL of 0.5% bupivacaine (Bupivacaine^®^; STERIMAX, Oakville, ON, Canada) was administered to achieve topical anesthesia.

The animals underwent transscleral retinopexy in the left eye, performed with an 810 nm diode laser (Iridex, Medical Oculight, Toronto, ON, Canada), using a laser spot size of 300 µm, power of 800 mW, and exposure time of 1000 ms, while the right eye was left untreated (control group). The laser probe (DioPexy Probe, Iridex, Medical Oculight, Toronto, ON, Canada) was in direct contact with the sclera through Tenon’s tissue and conjunctiva during the procedure. The posterior surface of the globe was divided into four quadrants: dorsomedial, dorsotemporal, ventromedial, and ventrotemporal. Three laser spots were applied to each quadrant on the posterior surface of the eye for a total of twelve laser spots. After completing transscleral retinopexy, topical ophthalmic ointment treatment with antibiotics and steroids (dexamethasone, polymyxin B sulfate, and neomycin sulfate; Maxitrol^®^; Novartis Pharma Inc., Québec, QC, Canada) was initiated twice daily for two days in the left eye. In addition, 0.03 mg∕kg IM of buprenorphine (Buprenex^®^; Renckitt Benckiser Pharmaceuticals, Richmond, VA, USA), a systemic analgesic, was also administered to all rabbits. Ophthalmic examination follow-ups from day 0 (D0) to day 42 (D42) included distance examination, ophthalmic evaluation, slit lamp biomicroscopy (Kowa SL-17^®^; Innova, Toronto, ON, Canada), and indirect ophthalmoscopy using a 3.5 V Finoff transilluminator (Heine, Topcon, Boisbriand, QC, Canada) and 30D and 20D lenses (Volk, Axis Medical, Anjou, QC, Canada). Ophthalmic examinations did not reveal any ocular inflammation two days after the transscleral retinopexy.

### 2.4. SD-OCT Scanning

OCT images of both eyes were acquired on day 0 (D0), day 1 (D1), day 7 (D7), day 15 (D15), day 21 (D21), and day 42 (D42), using the same sedation procedure as previously described. An eyelid speculum was used to keep the eyes open, and any corneal desiccation occurring during OCT scanning was prevented by maintaining corneal hydration using a topical gel containing 0.25% viscoadaptive hyaluronan (Hi-drop Vet-plus^®^; I-Med Pharma, Montreal, QC, Canada). The rabbits were placed in sternal recumbency to acquire B-scan spectral-domain OCT (SD-OCT; Spectralis Heidelberg Engineering, Innova, Toronto, ON, Canada) images [17]. The following scans were obtained from the peripapillary area: superotemporal, ventrotemporal, superomedial, and ventromedial. Additional horizontal volume scan images of hypo- or hyper-reflective retinal lesions were acquired based on fundoscopic evidence of additional retinal lesions. To allow for a direct comparison of retinal layer thickness measurements over time, images were acquired from the same retinal lesion, using automated eye-tracking software (TruTrack; Heidelberg Engineering, Heidelberg, Germany). This automatic follow-up scan placement software with minor manual adjustments was used to ensure the same precise regions were imaged each time. A volume scan mode was used to obtain cross-sectional retinal images and to obtain volumetric retinal scans comprising 61 single lines in 9 frames. A 55° widefield OCT lens was used to examine and image the retinas. Four measurements were recorded from the retinal photocoagulation lesions in the OCT scan images: the thickness on the left side (LS), the thickness at the center (C), the thickness on the right side (RS), and the widest horizontal length (HL) of the lesion.

### 2.5. Histological Sections

Two rabbits (n = 2 rabbits per time point) were euthanized under deep sedation (see protocol above) by administering an intracardiac injection of 1.5 mL pentobarbital (Euthanyl^®^ 240 mg/mL, BIMEDA-MTC ANIMAL HEALTH INC., Cambridge, ON, Canada) on D1, D7, D21, and D42, to perform a postmortem enucleation and compare the OCT images by the histological sections.

After enucleation, OCT was performed to allow the globe to be placed correctly in the space, by visualizing on the OCT screen the myelin from the optic nerve head (ONH) in the horizontal plane. Then, a colored line was drawn on the posterior globe to mark the myelin horizontal plane, thereby establishing the histological section plane (Figure 1A). The eyes were fixed in Davidson’s solution and 70% ethanol for 24 h each. The lesions were identified before immersion in paraffin to estimate the distance required to obtain the lesions on the cut slides. Tissue sample slides were cut (4 µM) and stained with hematoxylin, eosin, phloxine, and saffron (HPS), and the histological lesions were qualitatively evaluated by two veterinarians, board-certified in ophthalmology or pathology. The small number of animals (n = 2 rabbits per time point) included in the study precluded the use of score-based descriptions. The following four measurements for the retinal photocoagulation lesions were acquired from the digitalized microscopy slides: the thickness on the left side (LS), the thickness at the center (C), the thickness on the right side (RS), and the widest horizontal length (HL) in the lesion.

### 2.6. Statistical Analysis

Numerical data were tested for normality by the Shapiro–Wilk test. Statistical comparisons between males and females were performed by an unpaired Mann–Whitney U test to confirm that the population enrolled in this study was homogeneous in terms of age and weight.

Linear mixed models (LMMs) were used to assess the effect of time (used as a discrete variable) on the four dependent variables (LS, C, and RS thicknesses, and HL). The powers of the different linear mixed models are presented, calculated with the simr package: right side LMM = 1.00 (95% IC = 0.96–1.00); central model = 1.00 (95% IC = 0.96–1.00); left side model = 1.00 (95% IC = 0.96–1.00); horizontal length model = 0.31 (95% IC = 0.22–0.41).

Since the data were paired, the ID of the individuals was used as a random factor in each model. Likelihood ratio tests (LRTs) were performed to obtain model outputs. Post hoc tests were performed twice if time had a statistically significant effect on the dependent variable in the model, or if the time variable had more than two modalities. Since there were multiple comparisons (between modalities on D0, D1, D7, D15, D21, and D42), the *p*-values for the post hoc tests were corrected using the Benjamini–Hochberg method. The reliability between the two methods (OCT scan images and histological tissue samples) for the four variables was calculated by intraclass coefficients (ICC), using parameters of two-way mixed effects, absolute agreement, and a single rater [18]. Statistical significance was set at *p* < 0.05. Statistical analyses were performed using standard statistical software (R version 4.0.3; R Foundation for Statistical Computing, Vienna, Austria).

## 3. Results

### OCT and Histological Photocoagulation Lesions Images

Ophthalmic examination of the left eye of all rabbits revealed mild conjunctival hyperemia on the day following transscleral retinopexy (D1), which disappeared the next day (D2). No clinical signs of ocular discomfort were observed after retinopexy.

Five to six retinal photocoagulation lesions and 12 laser spots were identified in each treated eye after histological processing. Not all lesions were detectable by OCT or *fundus* examination; however, at least three retinal photocoagulation lesions were examined and followed up with OCT in each rabbit. The shape of the lesions observed via retinoscopy varied between round and oval, and did not change over time (Figure 1B,C).

The retinal lesions visible on the OCT images were mostly located around ONH in the superior and inferior quadrants. The progression of each lesion over time was followed up with OCT (Figure 2A–F). The retinal lesion was characterized by focal retinal edema, bullous retinal detachment, and irregularity in the underlying choroidal layer on D0 (Figure 2A). Focal cystoid retinal edema characterized by the presence of intraretinal cysts was observed directly beyond the laser area on D1 (Figure 2B). The thickness and width of the photocoagulation lesions gradually decreased from D7 onwards, with no significant differences on D15 (Figure 2C,D). The retina was not visible on D21 and D42, and the lesion appeared excavated with RPE hyper-reflectivity and loss of choroidal structure (Figure 2E,F).

Significant differences were observed between the RS, C, and LS thicknesses (LMM: *p* < 0.001) over time. However, HL did not vary significantly over time (*p* = 0.584) (Figure 3). The RS, C, and LS thicknesses were significantly affected by time and presented a similar pattern; there was no significant difference between D0 and D1 (*p* > 0.05) (Figure 3). However, the three measures of thickness decreased significantly between D1 and D7 (*p* < 0.001) and were significantly lower on D7, D15, D21, and D42 than on D0 or D1 (*p* < 0.001 for all comparisons) (Figure 3). No significant differences were observed between D7, D15, D21, and D42 (*p* > 0.05) (Figure 3).

Results of post-hoc tests, comparing the different modalities of the time variable were evaluated. P values were corrected by the Benjamini-Hochberg method in Appendix A.

A decrease in the thicknesses (microns mean values and percentage) in the RS, C, and LS was observed over time (Table 1).

According to the reliability scale by Koo and Li (2016) [18], the reliability between OCT and histology was excellent for the measurement of C (ICC = 0.91, *p* < 0.001) and moderate for the measurement of RS (ICC = 0.72, *p* = 0.006). However, the reliability between the two methods was not significant for the measurements of LS and HL (*p*-value = 0.055 and 0.500, respectively).

The clinical OCT findings correlated with the histological changes at each time point, when possible (Figure 4A–H).

Ganglion cell edema and interruption of RPE were observed in the histological sections and on the OCT images on D1. However, even if the collapse by the external plexiform and nuclear layers could be visualized by OCT and histology, details of the morphological cell damage could only be observed in the histological sections. (Figure 4A,B) Moreover, only the histological sections revealed the interruption in the choroid and RPE, RPE hyperpigmentation, the disappearance of the outer nuclear and photoceptor layers, and the reduction in the internal and external plexiform layers in some lesions (Figure 4C). Increased hyper-reflectivity of all retinal layers was observed in the OCT images, without an obvious decrease in the retinal thickness (Figure 4D).

A reduction in the neuroretinal thickness was evident in the histological and OCT images on D21. Histological images revealed disruptions in all neuroretinal layers (Figure 4E), which corresponded to an area of hyper-reflectivity in the RPE, neuroretina, and choroid on the OCT image (Figure 4F).

The retina was thin at the lesion site and melanin migration with RPE hyperpigmentation was visible in the histological section on D42, which corresponded to shine enhancement in the OCT image (Figure 4G). The neuroretina was disorganized at the center of the photocoagulation lesion in the histological sections (Figure 4H).

No complications were encountered after transscleral retinopexy, except for a focal choroidal hemorrhage of 1 mm in the eye of one rabbit.

## 4. Discussion

The results of the present study revealed a significant decrease in the thickness of the photocoagulated neuroretinal lesions between D1 and D7. The central part of the retina was affected most significantly, presenting the greatest decrease in thickness over time, as observed by OCT and histological images. As reported by Benner et al., a complete chronic photocoagulation lesion was observed seven days after the laser procedure in our study [16]. The photocoagulation lesions, characterized by neuroretinal edema due to the laser burn, progressed to the formation of a visible scar, which could be assessed by OCT and histology. Photocoagulation lesions are atrophic scars that anchor to the neuroretina and prevent retinal detachment.

The central part of the lesion showed the highest reliability in the correlation between OCT and histological measurements compared to the RS and LS of the retinal lesion. This finding is not unexpected because lesions in the center are caused by the direct effect of the laser energy on the retina, resulting in a photocoagulation lesion with low variability in thickness. However, energy diffusion and absorption occur at the periphery of this lesion, depending on whether the positioning of the probe is on the outer part of the eye, which could explain the higher variability in measurements.

The pressure exerted on the sclera by the laser probe may have been insufficient to induce focal burns after each pulse. The DioPexy probe used in this study was specifically designed for transscleral retinopexy photocoagulation. It delivers laser light at an angle of 90° degrees to the probe shaft [2]. It is likely that due to the short distance between the scleral surface and the retina, very slight changes in the beam angle may greatly affect the intensity of the laser spot [19]. Thus, it is important to position the probe tip perpendicular to the sclera when indenting the sclera to maximize scleral transmission; in addition, the aiming beam must be perfectly focused [19]. The energy transmitted through the sclera can change depending on the degree of scleral indentation and could result in a different shape (round or oval) or severity of the photocoagulation lesion. Scleral indentation increases the passage of laser energy [20,21]. Inter-individual variability in the amount of melanin present in the choroid and RPE layer could also explain some of the variations in the size and severity of photocoagulation damage [2].

HLs in the photocoagulation lesions did not vary significantly over time. Thus, it can be concluded that the surface size of the lesion did not change with the chronicity of the healing process and that the initial damage resulted in the formation of an atrophic scar of a similar size.

Laser photocoagulation is a type of photothermal therapy that has been validated and accepted for the treatment of various retinal diseases [22]. Although its mechanism of action is not completely understood, the final result is the formation of a visible retinal scar [22]. Intraretinal burns that cause chorioretinal scars progress such that there are expanding areas of atrophy over time [22]. Recent hypotheses suggest that its therapeutic benefits are derived from the adjacent areas affected by a lower, sub-lethal, photo-thermal action and not from the “burned” area of photocoagulation necrosis [22]. These hypotheses suggest that equally effective laser therapy could be administered for the purpose of creating only non-lethal photo-thermal elevations with no intraoperative visible endpoint and minimum intensity photocoagulation (MIP) using laser protocols [22]. Moreover, the fact that sub-threshold MIP protocols can be as effective as more damaging laser treatments is encouraging and supports the hypothesis that the mechanism of action of photocoagulation, with or without visible burn and/or RPE alteration, is based on laser-induced modifications of endogenous gene expressions in the retina [23].

Since the OCT reflective responses and light microscopy used for histology are vastly different approaches for viewing cellular elements, the two techniques reveal varying complementary information regarding the lesions. There were focal sites of intense, high reflectivity from OCT, which corresponded to centrally damaged photoreceptors in the lesion. However, light microscopy demonstrated only mild morphological changes in the damaged photoreceptors on D1. OCT identified changes in tissue reflectivity that could not be measured using light microscopy. However, the increase in hyper-reflectivity by the photocoagulation lesion over time in the OCT images later corresponded to hyperpigmentation and melanin mobilization in the histological section, which was visualized clearly on D42.

This study had some limitations, such as the small number of photocoagulation lesions detected by OCT and observed in each eye. Even if OCT and laser retinopexy were performed by the same person, a board-certified veterinary ophthalmologist with experience in this procedure, perhaps an incomplete laser retinopexy probe application on the scleral surface might prevent the production of a complete retinal lesion. Previous literature also reports that the absorption of energy in the sclera and pigments in the choroid could function, at least theoretically, as a barrier to the passage of the intraretinal diode laser toward the retina [3]. The small number of photocoagulation lesions did not allow for a severity score-based statistical analysis of the photocoagulation lesions. However, the main goal of this study was to characterize the progression of each photocoagulation lesion over time. In future studies, specific stains (Masson’s trichrome and reticulin) or immunohistochemistry (vimentin) can be used to determine the potential role of fibroblasts or gliosis in these lesions. This may also aid in determining whether the development of an epiretinal membrane (ERM) occurs secondary to repair processes, such as gliosis and fibrosis, which underlie cell contraction. Unlike photocoagulation, this type of scar can cause RD in humans [24]. Gliosis of neurological tissue develops from Müller’s cells and astrocytes, whereas fibrosis occurs following the proliferation of RPE and fibroblasts. The production of fibroblasts can result from an adaptive modification of RPE and hyalocytes [24]. The initial coagulation of the retina due to the energy delivered by the laser, followed by secondary gliosis, explains the adhesive effect of the lesions [19]. In the present study, a decrease in the thickness of the photocoagulation lesions over time confirmed the development of a chorioretinal atrophic scar, which will ensure the attachment of the retina to the choroid. Gliosis and fibrosis were not observed in the present study. The results of the present study highlight the potential of transscleral diode laser therapy for the treatment of retinal surgical conditions and the ability of OCT to detect the progression of photocoagulation lesions over time. Another limitation is the lack of significance in the reliability between OCT and histology, for LS and HL. The authors cannot explain the lack of significance in LS reliability, but regarding LH, it is possible that OCT and histological cross-sections are not exactly on the same plane. However, as reported by Koinzer et al., OCT images indicate different tissue changes than histologic images. After retinal photocoagulation, they show wider horizontal damage diameters but underestimate axial damage, particularly during healing. Thus, conclusions on retinal restoration should not be drawn from OCT findings alone [25].

The histopathologic consequences of laser photocoagulation have been studied, and the immediate effects noted were edema, inflammation, and necrosis, resulting from tissue damage, which had appeared to settle considerably 3 to 4 days after laser treatment [26,27]. The retinal lesions obtained in our study were mostly characterized by grade 2 burns, as described by Wollow et al. The retinal repair was characterized by Müller cell hypertrophy, the proliferation of pigment epithelial cell layers across Bruch’s membrane, and without severe choroidal and scleral injuries, as reported in grade 3 burns [16]. A grade 2 burn was considered to yield a successful and effective retinopexy with limited ocular inflammation and ocular blood breakdown [28,29,30].

Further studies would aid in the evaluation of the impact of the tensile forces exerted on the neuroretinal tissue, which is attached to the retinal scar produced by photocoagulation, to assess their efficacy in preventing or limiting future detachments.

## 5. Conclusions

The progression of photocoagulation damage from edema and resorption of the damaged retinal tissue (coagulative necrosis) to atrophic scar formation in rabbit retinas was studied over time using OCT and histological images in the present study. Based on the comparison between these two modalities, retinal edema, and RPE hyperpigmentation on histological images corresponded to increased hyper-reflectivity in the OCT images. Moreover, retinal thickness varies depending on the chronicity of the lesions.

SD-OCT is an accurate and useful tool for the identification of early retinal changes and monitoring retinal damage over time. Furthermore, it could be useful in assessing the effectiveness of treatment with retinal photocoagulation. To the best of our knowledge, this is the first study to obtain OCT images of photocoagulation lesions in rabbits that underwent transscleral retinopexy using an 810 nm transscleral diode laser at 1000 W for 5000 ms.

## Figures and Tables

**Figure 1 vetsci-10-00535-f001:**
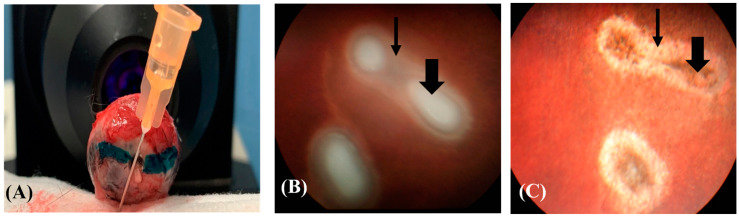
Rabbit globe. (**A**) A green-colored line is drawn parallel to the myelin bands observed in the OCT images to orient the histological section plane. (**B**) The photocoagulation lesion is surrounded by retinal oedemas (thin arrow) and central coagulative necrosis (thick arrow) on D0. (**C**) The same lesion has progressed to show central pigmentation (thick arrow) and a peripheral white ring (thin arrow) on D7 (Kowa Genesis-D Retinal camera, Innova, Toronto, ON, Canada).

**Figure 2 vetsci-10-00535-f002:**
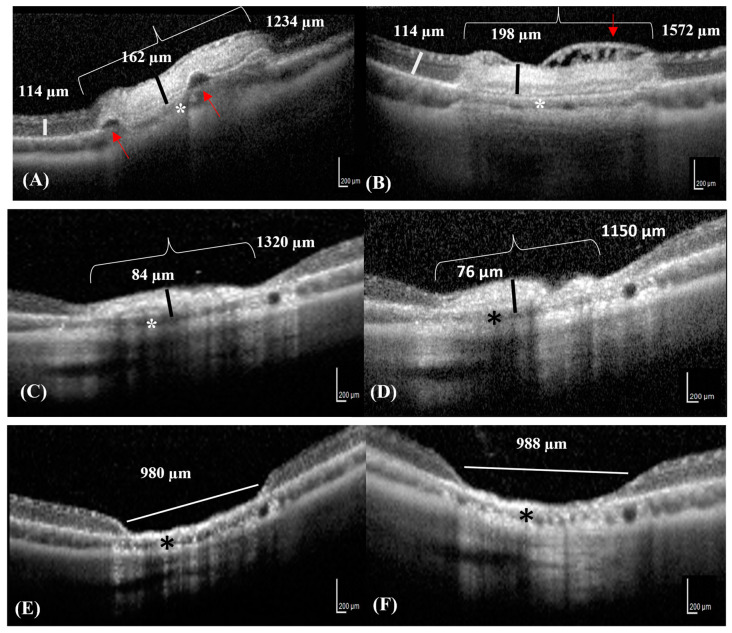
SD-OCT images of the same lesion progressing over time in a rabbit. (**A**) The OCT image reveals focal serous retinal detachments (red arrows) in an area of neuroretinal edema (bracket) on D0. The thickness of the neuroretinal edema (black line) contrasts with that of the adjacent healthy neuroretinal tissues (white line). (**B**) The OCT image reveals diffuse retinal edema with small superficial nerve fiber bullae, appearing as hypo-reflective intraretinal cysts (cystoid retinal edema) (red arrow) on D1. (**C**,**D**) The photocoagulation lesion appears hyper-reflective with decreased neuroretinal thickness and resolving retinal edema (black lines) on D7 and D15, respectively. (**E**,**F**) The retina Is completely atrophic and reduced to 1/10th of its original thickness (white line) on D21 and D42. (**A**–**F**) The choroid is slightly reduced in thickness in all Figures (asterisks).

**Figure 3 vetsci-10-00535-f003:**
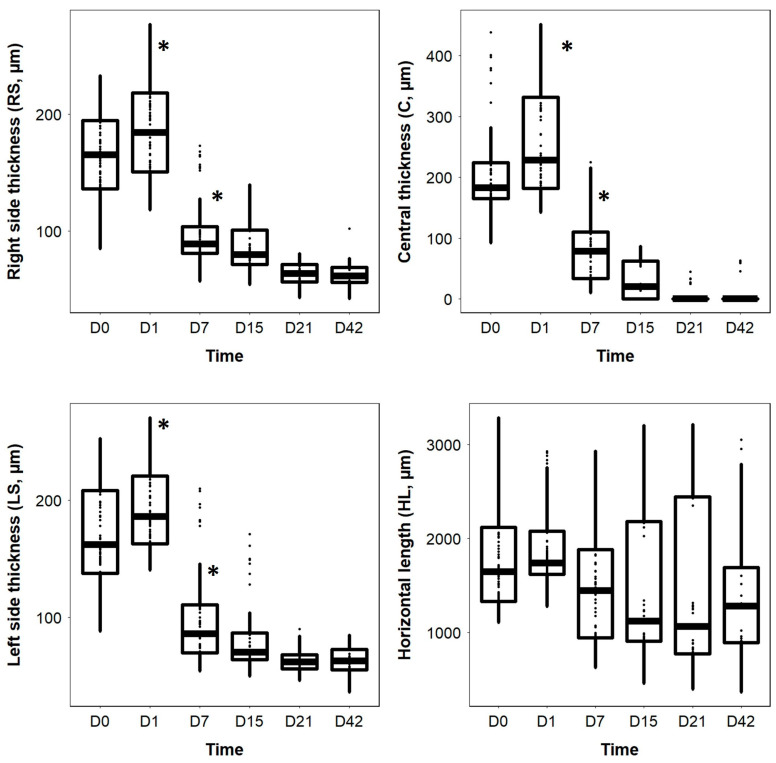
Boxplots of the right side thickness (RS), central thickness (C), left side thickness (LS), and horizontal length (HL) of the surface retinal lesions as a function of time. Asterisks * represent a statistically significant difference.

**Figure 4 vetsci-10-00535-f004:**
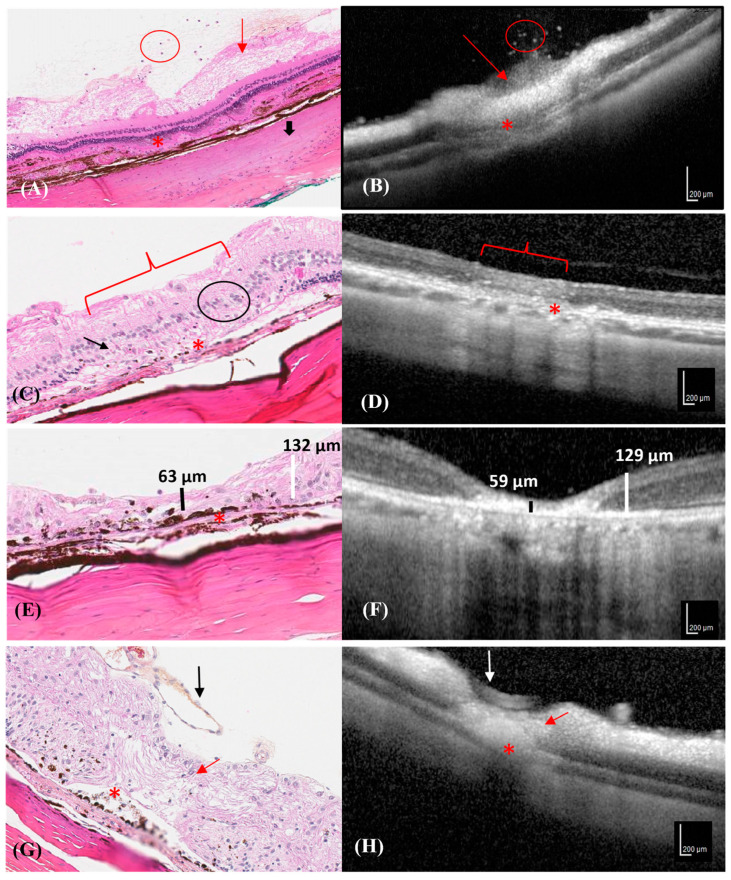
Progression of different photocoagulation lesions in the histological and OCT images of a rabbit. Hematoxylin–eosin stained retinal section corresponding to the area depicted by the OCT image. (**A**) Neutrophilic cells in the vitreous (red circle), ganglion cell edema (red arrow), RPE interruption (asterisk), and focal scleral hyalinization (thick black arrow), were observed in the histological section on D1. (**B**) SD-OCT transversal scan shows hyper-reflective dots in the vitreous (red circle), a hyper-reflective area characterized by ganglion cell edema (red arrow), and RPE interruption/choroid collapse (asterisk) on D1. (**C**) The histological section shows the disappearance of the outer nuclear layer (black arrow), the reduction in the internal and external plexiform layers (black circle), the disappearance of the photoceptor layer (bracket), and RPE and choroidal interruption with RPE hyperpigmentation (asterisk) on D7. (**D**) SD-OCT transversal section shows mild hyper-reflectivity of all neuroretinal layers (bracket), and RPE and choroidal interruption (asterisk) on D7. (**E**) A histological section showing a thin damaged neuroretina (vertical black line) compared to the adjacent healthy neuroretina (vertical white line) and RPE hyperpigmentation and disruption line (asterisk) on D21. (**F**) SD-OCT section shows a thin, hyper-reflective neuroretina (vertical black line) compared to the adjacent healthy retina (vertical white line) on D21. (**G**) A histological section shows a thin and disorganized neuroretina (red arrow), RPE hyperpigmentation with melanin migration (asterisk), and a prominent normal retinal vessel (black arrow) on D42. (**H**) SD-OCT section shows neuroretinal hyper-reflectivity, retinal thinning (red arrow), RPE interruption/choroid collapse (asterisk), and a normal retinal vessel (white arrow) on D42.

**Table 1 vetsci-10-00535-t001:** Values from the retinal thickness (micron) and changes (in percent) in the retinal thickness and length of the surface lesion over time. Asterisks * represent a statistically significant difference.

Time	Right Side (RS) Mean (±SD) %	Central (C) % Mean (±SD) %	Left Side (LS) % Mean (±SD) %	Horizontal Length (HL) % Mean (±SD) %
D0	164.3 (38.1) 0.00	200.9 (69.2) 0.00	169.9 (45.2) 0.00	1841 (630) 0.00
D1	188.2 (41.0) + 14.19	258.3 (87.1) + 28.51	194.2 (36.5) + 14.55	1906 (454) + 3.51
D7	98.3 (32.0) * − 49.15 *	88.9 (65.7) * − 65.59 *	98.7 (40.7) − 47.75 *	1461 (622) − 23.33
D15	87.0 (21.2) − 15.49	30.4 (31.8) − 65.83	83.4 (31.4) − 11.58	1455 (821) − 0.45
D21	63.8(10.1) − 24.27	5.5 (12.8) − 81.89	63.1 (10.3) − 26.59	1484 (991) + 2.06
D42	62.6 (11.3) + 1.61	12.9 (24.2) + 134.27	(64.2) (12.9) − 1.97	1357 (837) − 8.59

## Data Availability

Not applicable.

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
