# Peer review of "Morphological Evaluation of Transscleral Laser Retinopexy in Rabbits: Comparison of Optical Coherence Tomography and Histologic Examinations"

_vetsci, 2023, doi:10.3390/vetsci10090535_

Round 1

Reviewer 1 Report

The article is very interesting, but in our opinion has to be corrected in many points.

We would like to see the corrections correlated to our advices and then we will decide if it could be approved or not.

Reviewer 2 Report

  Congratulations for the work - the work makes a comparison related to transcleral retinopexy as a preventive technique against retinal detachment. Some short comments, aspects that can be improved: line 92: (120 × 100 × 100 cm) = L x W x H? Line 100: more details about the study - prospective, descriptive study... which? Line 181 - Linear Mixed Models Was it significant in 8 repeated measures? What was the estimated power in the linear mixed models? In the results section: We would like to see if there was homogeneity in terms of age and weight (non-parametric test results). For 4 males and 4 females. Was variability expected between sexes? line 277 of Table 1 you must write the values of Do measurements (as in the figures above). Below 0.00 you can put the values 0.00 ( ...micron). The discussion chapter - more structured and richer in citations/comparisons.

Reviewer 3 Report

In this manuscript, authors have described the qualitative and quantitative assessment of the photocoagulation changes induced in rabbits over the course of six weeks after undergoing 810 nm transscleral diode laser retinopexy and compare the OCT over ages with histological sections. This is a small interesting study, and the manuscript is generally written well. However, authors are advised to revise the manuscript based on the following comments.

1. Authors only showed the OCT-Bscans of the lesion which is measured in length/thickness. Could authors show the 3D image of the lesion from the OCT volume data and provide a 3D measurement or volume of the lesion? This would provide a better quantitative assessment.

2. It would be better to provide a general comment on OCT imaging technology (in intro) that it as a non-invasive in vivo optical imaging modality capable of providing cross-sectional and volumetric images of the retina of human and veterinary species [1-2]. (1) https://doi.org/10.1167/tvst.11.8.11    (2) https://doi.org/10.1111/j.1463-5224.2012.01045.x

3. Authors need to show the statitical test results in the Figure 3.

4.  Can authors provide a photograph of the imaging done invivo on rabbits?

A proof reading of the complete manuscript needed to be done to correct spelling and sentence errors.

Round 2

Reviewer 1 Report

I have no other comments to add

Author Response

Thank you for your suggestions and revisions